# Does (mis)communication mitigate the upshot of diversity?

**Keith Hankins**[1]*, **Ryan Muldoon**[2], **Alexander Schaefer**[3]

**1** Smith Institute for Political Economy and Philosophy, Chapman University, Orange, California, United States of America, **2** Department of Philosophy, University at Buffalo, Buffalo, New York, United States of America, **3** Classical Liberal Institute, New York University, New York, New York, United States of America

* keith.s.hankins@gmail.com

## Abstract

This paper contributes to the literature on how diversity impacts groups by exploring how communication mediates the ability of diverse individuals to work together. To do so we incorporate a communication channel into a representative model of problem-solving by teams of diverse agents that provides the foundations for one of the most widely cited analytical results in the literature on diversity and team performance: the "Diversity Trumps Ability Theorem". We extend the model to account for the fact that communication between agents is a necessary feature of team problem-solving, and we introduce the possibility that this communication occurs with error, and that this error might sometimes be correlated with how different agents are from one another. Accounting for communication does not give us reason to reject the claim associated with the theorem, that functionally diverse teams tend to outperform more homogeneous teams (even when the homogeneous teams are comprised of individuals with more task relevant expertise). However, incorporating communication into our model clarifies the role that four factors play in moderating the extent to which teams capture the benefits of functional diversity: i) the complexity of the problem, ii) the number of available approaches to solving the problem, iii) the ways of encoding or conceptualizing a problem, and iv) institutional characteristics, such as how teams work together. Specifically, we find that whether (and to what extent) teams capture the benefits of functional diversity depends on how these four factors interact with one another. Particularly important is the role institutional dynamics (like search methods) play in moderating or amplifying interpersonal frictions (like miscommunication), and notably we find that institutions that work in one setting can be counterproductive in other settings.

## 1. Introduction

The benefits and costs of diversity have each been the subject of a great deal of research in the social sciences. The role diversity plays in contributing to problem solving in groups [1, 2], spurring innovation [3–5], reducing organizational vulnerability to risk [6–8], improving the performance of firms [9–12], and increasing our ability to accurately diagnose illness have all received significant attention, and appeals to the benefits of diversity have also come to play an

**Data Availability Statement:** All relevant data are within the manuscript and its Supporting information files.

**Funding:** The author(s) received no specific funding for this work.

**Competing interests:** The authors have declared that no competing interests exist.

increasingly important role in justifications of democratic institutions [13, 14]. While, on the cost side, the fact that diverse groups tend to be less cohesive and experience more conflict are well documented [15–17], and the role diversity plays in contributing to misunderstanding, miscommunication, and the misalignment of goals and values have all been posited as mechanisms for explaining this.

However, the relationships between the costs and benefits of diversity are still not well understood, and the two halves of the diversity literature have not built off each other as much as they could. In particular, although there has been a lot of empirical work exploring what the impacts of various types of diversity are in different contexts, and a fair amount of theoretical work suggesting that the aspects of diversity that account for its benefits are also responsible for its costs [18–22], attempts to model the impact of diversity on groups have tended to focus on either its costs or its benefits while holding the other side of the equation fixed. This is unfortunate for two reasons. First, leveraging the benefits of diversity provides one path for addressing the costs and challenges associated with it, and attempts to leverage these benefits are more likely to succeed if they account for the costs associated with diversity. Second, if the theoretical literature is right that the mechanisms that account for diversity's costs are often what create the channels through which its benefits are derived, then successfully capturing the benefits of diversity depends not just on accounting for its costs, but on understanding these mechanisms. For instance, lack of trust within a diverse group might redound to the benefit of a group when it causes the members to raise the epistemic standards they hold one another to, but this will not always be true. Similarly, attempts to resolve disagreements stemming from diverse perspectives on an issue might generate new insights, but sometimes the disagreements can't (or won't) be overcome.

This paper explores an important aspect of the relationship between the costs and benefits of diversity that stems from the role communication plays in mediating the ability of individuals to work together. To do so we incorporate a communication channel into a model of problem-solving by teams of diverse agents that is representative of a broad class of problems that actual individuals and organizations confront, and that is associated with one of the most widely cited analytical results in the literature on diversity and team performance: the "Diversity Trumps Ability" (DTA) Theorem [2]. Specifically, we extend the model to account for the fact that communication between agents is a necessary feature of team problem-solving, and we introduce the possibility that this communication occurs with error, and that this error is correlated with various measures of how different agents are from one another. Accounting for communication does not give us reason to reject the increasingly widely held view, associated with the DTA theorem, that functionally diverse teams tend to outperform more homogeneous teams (even when the homogeneous teams are comprised of individuals with more task relevant expertise). However, incorporating communication into our model clarifies the role that four factors play in moderating the extent to which teams capture the benefits of functional diversity: i) the complexity of the problem, ii) the number of available approaches to solving the problem, iii) the ways of encoding or conceptualizing a problem, and iv) institutional characteristics, such as how teams work together. More importantly, our work suggests that whether (and to what extent) teams capture the benefits of functional diversity depends on how these four factors interact with one another, and we find that these interactions are often surprising. Two findings are particularly noteworthy. First, team dynamics that serve, in some contexts, to mitigate the costs of miscommunication associated with perspectival diversity will sometimes magnify those costs in other contexts. Indeed, the relationship between the four factors identified above is often non-linear and sometimes even non-monotonic. Second, in certain contexts the presence of miscommunication which we would typically assume to be costly may actually be beneficial.

We suggest that these results highlight the need for more work examining the ways that institutional arrangements interact with interpersonal dynamics to shape the effects of diversity within organizations. While our work focuses on a single interpersonal dynamic, namely communication frictions, many of the lessons we draw are generalizable to other such dynamics. Subsection 1.1 of the paper provides an overview of the DTA theorem and related analytical results about the benefits of diversity. Subsection 1.2 describes the need for incorporating communication channels that include the possibility of miscommunication into our models of diverse teams. Section 2 describes the model we explore. Section 3 summarizes the results of our analysis of this model. Section 4 provides our interpretation of these results and highlights several important takeaways. Section 5 concludes by discussing some of the limitations of our model and sketches some directions for further research.

## 1.1 Rugged landscape optimization, the diversity trumps ability theorem, and the Hong-Page model of diversity

The DTA theorem is an analytical result complemented by a set of computational experiments that characterizes a range of conditions for which diverse groups of problem solvers will tend to outperform groups composed of experts. The theorem is proven for a class of "rugged landscape hill climbing problems" that provide an illustrative way of modeling optimization problems common in many domains. Models like this represent problems as $n$-dimensional landscapes. One dimension constitutes a value function that reflects how good various solutions to the problem are relative to one another. That dimension is a function of the $n-1$ other dimensions that represent the features that distinguish various approaches to solving the problem. Points on the landscape are thus representative of specific approaches to solving a problem, and the model's object of analysis is how individual agents, or groups of agents, ought to go about exploring the landscape if their goal is to maximize the value function, where the size and ruggedness of a landscape represents the complexity of the problem, the resources that might be brought to bear on it, and the variety of approaches that might be taken to solve it. The rugged landscape hill climbing label for models like this derives from two or three-dimensional representations of problems that generate landscapes characterized by peaks and valleys, where maximizing the value function corresponds with identifying the highest peak on the landscape. In the version of the model associated with the DTA theorem the object of investigation is how diversity impacts the ability of groups to collectively identify solutions to optimization problems in cases where agents have common value functions (that is, where agents are assumed to agree on the relative merits of any given solution). Diversity is operationalized by considering agents who differ in their approaches to exploring the landscape. These approaches are defined in terms of fixed and finite heuristics that govern how the agents move about the landscape. Because points on the landscape represent solutions to the optimization problem being modeled search heuristics characterize which potential solutions to the problem an agent will investigate given the previous solutions she has considered. Search is therefore path-dependent, and the fixed and finite nature of search heuristics reflects the assumption that agents engaged in solving optimization problems will typically possess some (but not all) of the skills potentially relevant to solving the problem and given their skillset and the resources made available to them each agent will approach the problem in a specific way.

We call the model described above the *Hong-Page model of diversity*. In the extension of that model we consider here the landscape itself consists of 2,000 positions laid out in a two-dimensional ring structure. This corresponds with the computational experiments that Hong and Page describe [2], and directly extends an agent-based model developed by Daniel Singer to facilitate further investigation of the Hong-Page model [23]. Each position on the ring is

correlated to (some of) its neighbors, where the degree of correlation can be taken as a proxy for the complexity of the solutions. Highly complex landscapes exhibit low correlation; they are "rugged," and thus exhibit many local optima distinct from any global optimum. The heuristics used to search on this landscape are defined as $n$-tuples which specify an iterative sequence of moves that an agent might make about a landscape in the attempt to identify its highest point. For example, the 3-tuple (1, 11, 4) defines a search strategy in which an agent first explores the location 1 unit to the right of where she presently is on the landscape. If that spot is higher she moves there, and then renews her search by exploring the location 11 units to the right of that. If that location is not higher, though, she returns to her original location and explores the location 11 units to its right. Then, if that spot is higher, she moves there and continues her search by exploring the location 4 units to the right of her new location. And if that spot is not higher she again returns to her original location and explores the location 4 units to its right. Finally, if that spot is higher, she moves and iterates her search procedure by exploring the location 1 unit to its right. While if that spot (like the two previous locations she explored) is not higher, she will have exhausted her search strategy and will determine that her present location is the best she can do.

Utilizing this framework, we can explore how the composition of groups of agents affects their ability to work together in exploring a landscape. The primary implication of the DTA theorem and the headline result of the agent-based models that illustrate it is that "diverse" groups, comprised of individuals who adopt different approaches to exploring a landscape, tend to outperform "expert" groups comprised of the agents who are individually best at identifying the highest peak on a given landscape. More specifically, if expertise is defined in terms of the past performance of individuals in similar landscapes, then groups comprised of random agents tend to perform better than groups comprised of experts because random groups tend to be more functionally diverse [2].

There are two key dynamics at play here: *functional diversity* and *expert homogeneity*. Functional diversity manifests itself in the tendency of diverse groups to explore more terrain. This dynamic is driven by the fact that, from any given starting location, a diverse group will tend to set off exploring in many different directions, whereas a more homogeneous group will explore fewer possibilities. Of course, many of the directions in which the members of a diverse group set out may bear little fruit, but because they adopt different approaches to search, members of a diverse group are also less likely to get stuck at the same time or in the same location. This means that diverse groups will be less likely to conclude their search at a local (as opposed to global) optimum, and, even when this isn't true, they will tend to have considered more local optima. On the other hand, expert homogeneity refers to the fact that, in contexts where expertise is defined with respect to an individual's ability to perform a task, groups of experts are likely to be relatively homogenous. In the case of hill climbing problems, this is because the set of optimal individual search strategies will tend to be similar to one another for all but the most rugged landscapes. Specifically, random groups of agents will tend to be more functionally diverse than groups of experts when: i) the landscape is sufficiently rugged, ii) there are a large number of possible search heuristics available to agents, and iii) groups of agents are sufficiently large without being too large [24]. To see why note that, in all but the most rugged landscapes, optimal individual strategies will include some component of very local search that ensures an agent climbs a hill if she is on one, complemented by a component of global search suited to the ruggedness of the terrain that allows the agent to jump to a new hill. For example, in a two-dimensional landscape of the sort explored in the computational experiments that we extend this local component might involve exploring one unit left and, if that isn't higher, one unit right, while the global component will need to be at least as large as the mean distance between peaks on the landscape. When individuals work together

in teams, though, the local components of these optimal search strategies become redundant. In other words, one way of thinking about the diversity trumps ability result is that the best teams will comprise the individuals with the most complementary search strategies, and in this context diversity is closely associated with complementarity.

Having described the broad contours of the Hong-Page model of diversity a few caveats and clarifications are in order. First, two-dimensional landscapes like the one described above present a fairly simple problem space. Such landscapes are rugged enough to illustrate the benefits of diversity, though, and more complex problems can be modeled by straightforwardly extending the model to landscapes which are larger, more rugged, or which have more dimensions. Indeed, the analytical theorem associated with the DTA result is not restricted to the two-dimensional landscapes for which computational results have been described [2].

Second, how we measure diversity matters because there are different dimensions along which heuristics can differ, and some of these may be more relevant than others to how complementary (or functionally diverse) a set of heuristics is [23, 25, 26]. For instance, two heuristics (or sets of heuristics) might differ with respect to the number of unique moves they include, or with respect to the order in which employ moves. We can refer to the former as the coverage (or C-diversity) of a set of heuristics, and, at least in the context of two-dimensional landscapes, it appears that coverage and not order explains most of the benefit of diversity [23]. Even C-diversity may be too coarse-grained to capture how different two heuristics are, though, especially in more rugged or higher-dimensional landscapes. Indeed, any single measure of diversity is likely to overstate how complementary some sets of purportedly maximally diverse heuristics are, and so we should be careful not to overinterpret any particular result about the relationship between diversity and outcomes of interest.

Third, how the composition of groups impacts search isn't the only thing we might investigate. We can also ask how search dynamics impact things. In their initial analysis of this model, Hong and Page show that the DTA result is robust to how groups search together [2]. In particular they consider two models of group search: a) a sequential/relay model where one agent does the best she can on the landscape and then passes things off to a fellow group member who picks up the search where the previous agent left off, with this process repeating until no agent can improve on the previous agent's search, and b) a simultaneous model where each member of the group does the best she can exploring the landscape and then reports the results of her exploration to the other group members, at which point each member in the group moves to the highest peak identified in the first round of search and begins searching anew until no agent is able to improve upon the result of the previous rounds search. That team search dynamics would prove inconsequential is surprising. Whether that result generalizes or is instead an artifact of the baseline model's lack of complexity is one of the primary questions motivating our interest in exploring extensions of the model. For instance, one drawback of the Hong-Page model is that it only considers contexts where search heuristics are fixed (and so insensitive to what other team members do). While models that allow for the possibility that individual agents utilize search strategies that condition on what other agents do have similarly highlighted the role that diversity plays in making team search more productive [27–29], the primary thrust of those models is that search dynamics, and the context in which search takes place both matter.

Our operating assumption in what follows is that the Hong-Page model of diversity provides a useful framework for exploring some of the ways in which diversity impacts team problem solving. Furthermore, with the above caveats in mind, we contend that when interpreted with sufficient modesty pared down specifications of the model are representative of more complex specifications. Indeed, it is helpful to think of the Hong-Page model as a variant of an *NK model*. The first iterations of these models were developed to model adaptation in

evolutionary biology [30, 31]. On that interpretation the model characterizes a fitness landscape where the object of analysis is how agents facing selective pressures ought to go about choosing traits when fitness is a complex function of the traits they adopt, but *NK* models have since been applied to problems in fields as diverse as management [32–34], condensed matter physics [35], political philosophy [36], and computer science [37], and within the literature on collective problem solving these models have been especially influential [29, 38–40]. *NK* models are characterized by two parameters that together define the complexity of a problem being modeled. *N* represents the number of distinct inputs that potentially characterize solutions to a problem, while *K* represents how these inputs are related to one another. If we think of a problem as a value function to be maximized, then *N* describes the number of independent variables that the value function ranges over, and *K* represents the number of ways in which these variables interact with one another. As problems become more complex–i.e. as N and K increase–the landscape characterized by these parameters becomes more "rugged" thus creating the possibility that agents will get stuck at local optima during their searches. While the landscape we employ in our model lacks the dimensionality of more complex *NK* models, its ruggedness can be interpreted in a similar way. In particular, we can generate two-dimensional landscapes that are rugged enough to make identifying global optima difficult for individual agents who are constrained in their ability to explore the landscape. The model we describe here in which agents utilize fixed heuristics to explore a rugged landscape along a single dimension thus has the virtue of providing an easily depicted and tractable way of modeling optimization problems characterized by three fairly ubiquitous features:

i) *Constrained search*–Agents are only capable of engaging in so many attempts at solving a problem, and these attempts will not typically exhaust the number of distinct approaches an agent might employ to try to solve a problem.

ii) *Path dependence*–Which potential solutions an agent considers are determined by the particular approach(es) to solving a problem they adopt and the resources they bring to bear on it.

iii) *Evaluative consensus*–Agents agree about what would count as solving the problem in question, or, at least, once a potential solution has been explored agents agree about how to evaluate its relative merits. This feature is considerably more restrictive than the first two. In many cases, as we entertain increasingly diverse perspectives on a problem, that diversity is likely to engender disagreement with respect to how to evaluate solutions to the problem [41]. Nevertheless, problems where the assumption of evaluative consensus is reasonable remain important and common in many domains.

## 1.2 Perspectival diversity and difficulties communicating

As we described above, the fact that diverse groups are able to outperform groups of experts on tasks that can be modeled as rugged landscape optimization problems is a function of the fact that groups of experts tend to be homogeneous. Because they will tend to adopt different perspectives on a problem or employ different approaches to solving it, diverse groups will explore more possibilities, and, as a result, are less likely to get stuck at local optima. The same functional aspects of diversity that generate this benefit, though, are also likely to impose costs. Many, although by no means all, of those costs are associated with difficulties communicating.

One obvious source of communication costs is the need for translation when individuals speak different languages, or simply use different shorthands. There are two types of cost associated with translation. The first stem from mistranslation. Mistranslation occurs both when

an agent makes a mistake in translating and when something is lost in translation (for instance, because there is not a one-to-one mapping between languages). Both make collaboration more difficult and less productive. For example, in sequential work where group members build upon each other's contributions, mistranslation might prevent one individual from picking up where the other left off. While in collaborative work that takes place simultaneously mistranslation can lead individuals to work at cross-purposes without necessarily realizing it. Second, translation also creates costs associated with the opportunity cost of time. Some of these costs stem from mistranslation. For instance, when group members work at cross purposes, or make mistakes in building upon each other's work, they might have to backtrack or repeat work. Alternatively, when mistranslation leads to disagreement that group members are aware of, they must spend time resolving the disagreement. Translation can also be costly even when there are not errors in translation. This is because the need for translation typically slows down the speed with which information can be shared, meaning that collaborative work takes longer than it otherwise would.

Furthermore, translation is not the only source of communication costs. While translation slows down the speed with which information can be shared, and introduces the possibility of mistranslation, both of these costs are associated with communication more generally. The process of sharing information takes time even when translation is not necessary, and any time information or instructions are passed from one agent to another there is the possibility that information will be lost, degraded, or transformed along the way.

Moreover, the need for translation does not merely arise when members of a group speak different languages. Rather, it is necessary any time agents represent the world in different ways, and, in addition to language, there are at least three other ways in which agents might represent the world differently [34, 41–43]. The first reflects differences in background knowledge and experience that an agent might bring to bear on a task/problem. The second, differences in the features of the world that an agent is attentive to, how they model a problem/task, and how they generate ideas or solutions for resolving the task. And the third, differences in their values and goals, including the weight or priority that they give to conflicting goals/values. When agents differ along some or all of these dimensions the need for translation and the costs of communication both increase, and this is compounded by the fact that the aspects of representation are related and so differences of one type often beget differences of another. Put another way, agents who are working together to solve a problem, but who think about the problem or approach solving it differently are likely to have to spend time explaining their approaches to one another. This is time that homogenous groups do not have to spend. If we conceive of a problem in the same way and agree on how to approach it we do not need to explain the details of our approach to one another, except perhaps to simply confirm that we are on the same page.

Furthermore, because the need for explanation introduces (or exacerbates) the possibility of miscommunication, and because some ways of representing the world will be more compatible with one another than others, there will be some groups of agents for whom miscommunication is more likely to occur (and some contexts in which this miscommunication is more likely). Indeed, at some point it may become impossible for agents with rival perspectives to effectively collaborate with one another [41, 42].

Interestingly, Hong and Page acknowledge the importance of this issue, concluding their discussion of the model we extend by noting:

> The current model ignores several important features, including communication and learning. Our perspective-heuristic framework could be used to provide microfoundations for communication costs. Problem solvers with nearly identical perspectives but diverse

heuristics should communicate with one another easily. But problem solvers with diverse perspectives may have trouble understanding solutions identified by other agents.

(Hong and Page 2004: 16389)

To our knowledge nobody has developed the model in this way. This paper seeks to fill that gap in the literature. In doing so we have two complementary aims: i) to improve our understanding of when the DTA theorem holds and how broad its implications are, and ii) to augment our understanding of the way in which communication mediates the impact of diversity on teams. Although other attempts have been made to model the way in which diversity, communication, and teamwork influence one another, none of these attempts utilize the Hong-Page model of diversity that the DTA theorem is built on. Given how influential the DTA theorem has been in the literature on diversity and team problem solving, our paper is motivated by the view that incorporating communication into the Hong-Page model provides an important contribution to the literature.

In adopting this approach our work complements recent work on what the DTA theorem can teach us [44, 45], by exploring ways in which the model can be productively extended. Adopting this approach also distinguishes our paper from Frigotto and Rossi (2012) who are motivated by otherwise similar aims [46]. Their objective is to integrate the literature on diversity and team performance with the literature on communication in teams. Like us, they acknowledge the significance of the Hong-Page model but are critical of its failure to incorporate communication. However, because they are critical of other aspects of the model–especially, the way Hong and Page measure diversity and the extent to which their model can capture incomplete/asymmetric information–– they introduce an alternative framework for modeling things. While we are sympathetic with some of these concerns, our view is that Frigotto and Rossi overstate the extent to which they undermine the utility of the Hong-Page framework. More importantly, introducing an alternative framework as Frigotto and Rossi do, prevents us from determining how accounting for communication affects interpretations of the DTA theorem. The extent to which we build upon the Hong-Page framework is not the only way in which our approach to modeling diversity and communication differs from Frigotto and Rossi's, though. Where we investigate the impact of information being lost or transformed when agents communicate, they explore how the intensity of communication between agents affects their ability to work together. Our respective approaches should thus be conceived of as complementing one another (we say more about how in section 5). Also noteworthy in this respect is work by Fontanari and Rodrigues [39] and Takahashi et al. [40]. Neither of those papers directly extends the Hong-Page framework, but both utilize NK landscapes to model problem-solving. Specifically, the former explores the interaction between landscape topology, communication, and team structure on performance but does not consider diversity, while the latter explores how diversity and incentive structures influence performance but does not incorporate communication. Our approach thus builds on these approaches by incorporating topology, diversity, communication, and team dynamics within a single model.

## 2. Extending the Hong-Page model to include miscommunication

The model we describe here is based on the model developed by Daniel Singer that provides a straightforward agent-based representation of the Hong-Page model by imposing a rugged landscape on a 2000 unit ring that agents explore using 3-tuple heuristics [23]. Singer's model has also been used in other analyses of the Hong-Page model [24, 47]. However, where both the base Hong-Page Model and Singer's representation of it assume that communication between agents working together is lossless and costless, we introduce the possibility of

miscommunication. We also consider the possibility that miscommunication and diversity by exploring a version of the model in which the likelihood of miscommunication increases as agents become increasingly different (that is, as the search heuristics they employ become more different).

Notably, this way of introducing communication costs into the Hong-Page model differs slightly from the approach suggested in the quotation we pointed to above. The 'perspective-heuristic framework' mentioned there refers to an alternative interpretation of the model Hong and Page introduce that operationalizes diversity by considering agents who differ in their perspectives on what that landscape looks like (that is agents who differ from one another with respect to their beliefs about which points in a landscape are close to which other points) [48]. Hong and Page argue that the DTA result also obtains when we conceive of agents differing with respect to their perspectives on a problem as opposed to their approaches to solving it. While we agree with them that communication difficulties are likely to be more significant among agents with different perspectives in comparison to agents with different heuristics, miscommunication is still likely to be significant for agents with different heuristics (or approaches to solving a problem) insofar as productive collaboration typically requires such agents to explain their approaches to each other. For instance, modern experiments in physics now often require collaboration between teams of hundreds of scientists, who might include theorists, engineers, experimentalists, programmers, and data analysts, among others, few of whom will complete understanding of what their collaborators do [49]. Pragmatically, it is also easier to illustrate and provide a measure of how different agents are from one another when we define agents in terms of heuristics as opposed to perspectives. And, at least for low-dimensional landscapes, when it comes to interpreting the model the differences between heuristics and perspectives are relatively insignificant (33). Accordingly, we extend the version of the model where agents differ in their approach to search for three reasons: i) extensions of that version are easier to operationalize, ii) it is intuitively easier to grasp, and iii) papers employing it are much more commonly cited in the literature.

The following description of our model follows the ODD (Overview, Design Concepts, Details) protocol as laid out by Grimm et al. [50].

## 2.1. Purpose

The central question this model has been designed to address is the following: how does (mis) communication affect the ability of diverse individuals to solve problems together? As noted in the introduction, there is ample evidence that diversity benefits groups searching for solutions to complex problems. There is also evidence that diversity makes it harder for groups to effectively work together, e.g. by making it more difficult for them to communicate. The relationship between these two things has received insufficient attention, and the goal of this model is to explore that relationship. In particular, our model is designed to explore the way in which team search strategies interact with communication frictions that arise between agents to mediate the impact of diversity on team performance. To do so we incorporate a noisy communication channel into the widely cited Hong-Page model of diversity. Doing so allows us to analyze whether introducing the possibility of miscommunication impacts the interpretation of the Diversity Trumps Ability Theorem associated with that model.

## 2.2 Entities and state variables

**2.2.1 Problem-solving landscape.** The model utilizes spatial structure to represent constrained optimization problems by depicting problems as landscapes defined as rings comprised of 2,000 discrete points. Each point in the landscape represents a potential solution to

the problem, with the "height" of a point serving as a proxy for the relative quality of the solutions. Landscapes can exhibit various degrees of ruggedness, a proxy for the complexity of the problem. The degree of ruggedness is dictated by a smoothing parameter, S. Specifically, ruggedness is imposed on the ring-shaped landscape by a function that populates the landscape by randomly assigning a height $y_0 \sim U(0,100)$ to an initial point $x_0$, and then using a random variable $Z \sim U(1,2S)$ to identify a second point $x_i = z_1$ whose height is again assigned a randomly selected value $y_i \sim U(0,100)$. The heights of all intermediate points on the landscape $(y_1, \ldots, y_{i-1})$ are then imposed by a linear function connecting the randomly chosen endpoints $(x_0, y_0)$ and $(x_i, y_i)$. The random variable Z is then used to identify a new point $x_j = x_i + z_2$ on the landscape with height $y_j \sim U(0,100)$, and the heights of the intermediate points $(y_{i+1}, \ldots, y_{j-1})$ are again imposed by a linear function connecting the endpoints $(x_i, y_i)$ and $(x_j, y_j)$. This process repeats itself until all 2000 points on the landscape have been filled in. A smoothing parameter with a lower value thus generates a more rugged landscape in which the height of any given point is correlated with fewer of its neighbors.

**2.2.2 Problem-solving agents and teams.** The object of analysis of the model is how groups of agents explore the landscape, where success is defined in terms of identifying the highest peaks in the landscape. Each agent is characterized by a search heuristic, i.e. an ordered 3-tuple that determines how the agent explores the landscape of solutions as detailed in 1.1 above. Individual agents are then grouped into teams that engage in collective search. We consider two types of teams. One type consists of "experts" defined as the agents who perform best when engaged in individual search (where performance is averaged across every starting position in the terrain). The other type of team consists of "randoms," or agents drawn at random from the entire population of agents. As we discussed in 1.1 above experts will tend to have similar heuristics, while randoms will exhibit greater diversity, and in the conditions specified by the DTA theorem random teams tend to outperform expert teams because of this diversity (provided that landscapes are moderately rugged). The goal of the model is to assess whether the result associated with the DTA theorem holds up when teamwork requires communication and miscommunication is possible. To make our exploration of this question more robust, the model allows us to vary several parameters related to the agents that populate the model and the way they work together. The first of these concerns the maximum value that the components of an agent's heuristic can take on. This parameter determines how much of the landscape an agent can explore in a single move. We considered parameter values ranging from 4 to 25. Following Singer (who follows Hong and Page) the results reported in section 3 are for the specification in which each component of an agent's heuristic could take on a maximum value of 20. This represents the case where agents are able to explore the closest 1% of the possible solution space at any given time, and analysis of preliminary runs of the model suggested that the results we report are representative of other parameterizations. The number of components in an agent's search heuristic (i.e. the size of $n$-tuple characterizing each agent) can also be varied. We restricted our attention to the case where heuristics are 3-tuples because prior investigation described by Hong and Page suggests these results are representative of parameterizations in which agents are characterized by larger $n$-tuples. This parameterization also presents the clearest model of cases where team problem-solving is advantageous because individuals are constrained in their ability to explore the solution space.

The size of the teams within which agents work can also be varied. The results reported here are for groups of 9 agents. This approximates the approach of Hong and Page who describe results for a simulation utilizing 10 agents. However, utilizing 9 agents allows us to explore additional team search strategies. Preliminary investigation again suggested that these results were representative of runs of the model in which groups were larger and smaller.

In addition to the size of teams, the model also allows us to vary aspects of how teams of agents interact. In particular, we consider three different team search dynamics, and several ways of modeling the error that communication between agents might introduce. These are described in more detail in 2.7.

**2.2.3 Two measures of diversity.** As we discussed in section 1, interpretations of the DTA theorem can be sensitive to how we measure diversity. In our case, we are also interested in the possibility that miscommunication is correlated with how diverse a team is. To accommodate these considerations we employ two distinct measures of diversity/similarity commonly utilized in information theory: the first order Minkowski distance and the Hamming distance. In this context, the first order Minkowski distance defines the similarity of two agents as the sum of the differences between the respective components of the vectors that define the heuristics they employ, i.e. for $H = (h_1, \ldots, h_n)$ and $G = (g_1, \ldots, g_n)$, $d_{man}(H, G) = \sum_{i=1}^{n} |h_i - g_i|$. This distance measure is often referred to as the Manhattan (or taxi cab metric), so called because it provides an intuitive measure of how far a taxi would have to travel to get between two points in a city with streets laid out in a grid. The landscape we employ has only one axis along which agents move, but for the purpose of comparing how similar two heuristics are we can think of the components of our heuristics as distinct dimensions analogous to the street axes along which a taxi might move. On the other hand, the Hamming distance provides a measure of the similarity of two agents by comparing the respective components of the heuristics they employ and measuring the dissimilarity between them in terms of the number of components that are different, i.e. for $H = (h_1, \ldots, h_n)$ and $G = (g_1, \ldots, g_n)$, $d_{ham}(H, G) = \sum_{i=1}^{n} v_i$, where $v_i = 0$ if $h_i = g_i$, and $v_i = 1$ if $h_i \neq g_i$.

According to both measures heuristics with less distance between them are more similar. However, utilizing both measures allows us to capture the fact that two search heuristics may be similar in some respects, but different in others. For instance, for the set of heuristics including $H_1 = (1, 2, 3)$, $H_2 = (1, 2, 4)$, $H_3 = (1, 2, 12)$, $H_4 = (4, 5, 6)$, the Hamming distance between $H_1$ and $H_2$ is the same as the Hamming distance between $H_1$ and $H_3$. Specifically, $d_{ham}(H_1, H_2) = 0 + 0 + 1 = 1 = d_{ham}(H_1, H_3)$. But the intuitive sense in which the latter pair is less similar is captured by the fact that the Manhattan distance between them is greater: $d_{mam}(H_1, H_2) = 0 + 0 + 1 = 1$, while $d_{ham}(H_1, H_3) = 0 + 0 + 9 = 9$. Analogously, the Manhattan distance between $H_1$ and $H_3$ is the same as the Manhattan distance between $H_1$ and $H_4$, but the Hamming distance of the latter is greater ($d_{ham}(H_1, H_4) = 3$). While Hamming distance is more course-grained than Manhattan distance, then, in the sense that the former can take on fewer values, the latter is not necessarily to be preferred in cases where the two measures deliver conflicting verdicts.

Given our interest in the DTA theorem, Hamming distance is also important because it is analogous to the measure of diversity that Hong and Page utilize in characterizing the theorem (it is analogous, but not identical because the measure Hong and Page utilize is reported as a value $d_{HP}(H_1, H_2) \in (0,1)$ corresponding to the percentage of components that match for any pair of heuristics. Hamming distance is also similar in some respects to the measure of C-diversity we described above. The key difference is that the Hamming distance measure we utilize captures the fact that search heuristics are ordered, where the C-diversity measure does not. And C-diversity captures the percentage of possible moves that a set of heuristics utilizes, while Hamming distance does not. As a result, pairs of heuristics with the same measure of C-diversity might have different Hamming distances, and vice versa. Although previous analysis has suggested that HP-diversity lacks much explanatory power once we control for the C-diversity of a group (30), we utilize the Hamming distance measure for three reasons: i) C-diversity provides a useful measure of how diverse a group of agents is, but it does not provide

a useful measure of how different any two agents are from one another, and when we model miscommunication as a function of diversity it is the latter measure that we need, ii) even if the differences in performance explained by HP-diversity in the original analysis of the model can be better explained by C-diversity, HP-diversity is a decent proxy for C-diversity, and iii) because the order in which group members make moves is likely to become increasingly important as the possibility of miscommunication is introduced.As we describe in 2.7 below, how a team searches influences how they (mis)communicate. Since we consider cases where miscommunication is a function of diversity, this means that choice of search procedure influences how we operationalize the two measures of diversity described above. We describe the details of this operationalization in 2.7.

## 2.3 Process overview and scheduling

As described above the model explores the impact of communication frictions on team problem-solving by depicting constrained optimization problems as a ring-shaped landscape characterized by peaks and valleys that teams of agents explore. Agents whose individual efforts to explore the landscape are characterized by deterministic search heuristics use team search procedures to coordinate their individual search efforts with one another. Coordination is facilitated by communication which is subject to error. We consider three team search procedures: a *relay* model where agents search sequentially, a *tournament* model where agents search simultaneously in rounds from the same starting location, with each round beginning at the highest point identified in the previous round, and a *hybrid* model where teams are broken into subgroups where search within a subgroup utilizes the relay model and search between subgroups utilizes the tournament model. We also consider five ways of modeling (mis)communication: (i) perfect communication where no error is introduced, (ii) fixed error that is a function of how diverse agents are, and three stochastic models where error is assumed to be (iii.a) normally distributed and a function of how diverse agents are, (iii.b) drawn from a Poisson distribution, and (iii.c) drawn from an exponential distribution. Combinations of search procedures and models of communication constitute submodels and are described in detail in 2.7 below. Simulations of these submodels are run subject to various parameterizations of the landscape and the agents that explore it. The object of analysis is how the interaction between search procedures and models of communication impacts the performance of teams. Specifically, we compare the (absolute and relative) performance of teams of "experts" with teams of "randoms."

Runs of the model begin by generating a landscape characterized by a smoothing parameter as described in 2.2.1. The set of agents capable of exploring the landscape is generated where these agents are characterized by their 3-tuple search heuristics and the number of unique agents in the set is a combinatorial function of the maximum value that the 3-tuple search heuristic components can take on. For a given landscape the "average competence" of every individual agent is then defined as the average height of the point at which their search would conclude for every possible starting location on the ring if that search were guided entirely by their individual search heuristic as described in 1.1. "Expert" teams are generated by identifying the nine agents with the highest average competence on the landscape. The performance of these teams is then compared with the performance of "random" teams comprised of nine randomly selected agents. In both cases team performance is defined as the average height of the point at which a team's search concludes for all possible starting locations. For each landscape this measure of performance is identified for each of the three search methods we consider given one of the two diversity measures we consider and three ways of modeling communication (perfect, fixed error, and one of the three stochastic distributions we consider). Runs of

**Table 1. Parameters and submodel components.**

| Parameter | Values |
|---|---|
| Search Method | *Relay*<br>*Tournament*<br>*Hybrid* |
| Miscommunication Type | *Perfect Communication*<br>*Fixed Error*<br>*Normally Distributed Error*<br>*Exponentially Distributed Error*<br>*Poisson Distributed Error* |
| Diversity Measure | *Minkowski (Manhattan)*<br>*Hamming* |
| Smoothing Factor | *4, 6, 8, 10, 12, 14, 16* |
| Max Heuristic Component | *5, 10, 15, 20* |

the model are therefore characterized by four parameters: max heuristic component value, landscape smoothing factor, diversity measure, and stochastic error distribution. Table 1 below summarizes these parameters and submodel components, while the way these parameters function within submodels is described in more detail in 2.7.

## 2.4 Design concepts

Several design concepts were considered in the construction of this model. The primary goal of the model is to explore the non-additive, epistemic features of diversity within solution-seeking teams. The enhanced performance of such teams is an emergent property insofar as it is not the sum of the abilities of individual agents within a team, but rather a group-level feature shaped by relations between the individuals that comprise the team. Our project is to determine (i) whether or not this emergent property holds when communication frictions are introduced, and (ii) what conditions enhance or diminish it. In particular, we are interested in how the emergent property is affected by team dynamics and the possibility that communication frictions are a function of the diversity of a team.

The objective of agents in our model is to identify peaks in the rugged landscape they are exploring. As we saw (sec. 1.1) the landscape provides a representation of a constrained optimization problem with points on the landscape representative of specific solutions to the problem. The heights of points in the landscape are correlated with their closest neighbors and correspond to the relative quality of solutions.

Learning and communication are two other important design concepts that our model incorporates. While the search heuristics of agents are taken as fixed, agents learn from their team members insofar as they are able to see (and move to) parts of the landscape that they would not otherwise be able to when those team members report the results of their search. This process of communication and learning occurs differently in each of the three team search methods we consider but is present in each. Communication and learning are also significantly implicated in the way communication frictions are introduced into the model. As we describe in more detail in 2.7 below, we consider two places in which miscommunication can occur (between agents in the relay model and at the end of a round of search in the tournament model) and four ways of modeling this miscommunication. The implications of these ways of inducing miscommunication are then compared to a baseline model with perfect communication.

How agents interact with one another is an important feature of collective problem-solving. We incorporate two forms of interaction (i) communication between agents and (ii) the team

search procedures that govern how agents utilize the information communicated by their teammates. Although other forms of interaction are no doubt important, we restrict our attention to these forms of interaction to better identify the impact that incorporating communication has on interpretations of the DTA theorem.

Stochasticity features in the model at various points. First, the landscape itself is randomly generated, subject to a smoothing parameter (sec. 2.2.1). Second, each agent's search heuristic is randomly generated, subject to a maximum step size (sec. 2.2.2). Finally, the distinctive contribution of our model, namely miscommunication, is modeled as a random error (sec. 2.7), and even the fixed model of error we consider that is generated by a deterministic function of group-level diversity exhibits stochasticity insofar as groups are constructed randomly.

Collectives are the unit of analysis in our model. As explained above, the focus of the model is on the emergent epistemic properties of teams engaged in collective problem-solving subject to various ways of characterizing the problem space and team dynamics. Specifically, our investigation is motivated by the assumption that individual agents are constrained in their ability to explore a problem space. Working together allows agents to overcome some of these constraints, but the effectiveness of a team plausibly depends on how it is organized and what kind of frictions mediate the interactions of team members. Accordingly, teams provide an intermediate level of organization between whole populations and individual agents that constitutes an appropriate level of investigation.

Although adaptation and prediction are often important characteristics of how agents solve problems our model does not incorporate these design concepts. Once again, we have excluded these elements from our design because incorporating them into our model of agent behavior would make it more difficult to identify what drives the results.

Observation and data collection occurs by running computational simulations of the model subject to various parameterizations. As described in 2.3, runs of the model are characterized by four parameters (max heuristic component value, landscape smoothing factor, diversity measure, and stochastic error distribution), while submodels are distinguished by the team search method, model of (mis)communication, and the diversity measure. The results reported here summarize 8400 runs of the model in which each way of parameterizing the model was run 50 times. In particular, our emphasis is on the 2100 runs of the model in which the max heuristic component is 20. Each run of the model generates data on the absolute and relative performance of "expert" and "random" teams for all possible combinations of team search procedure, model of (mis)communication, and measure of diversity. This data was primarily analyzed using paired t-tests to identify statistically significant differences in the performance of these types of teams in the settings we consider.

## 2.5 Initialization

Various steps occur prior to collecting data on the performance of teams: constructing the landscape, generating the set of agents characterized by the search heuristics that are possible for a given parameterization, determining each agent's individual competence, and creating the"expert" and "random" teams whose performance will be compared. Each of these steps is independently undertaken for every run of the model.

## 2.6 Input data

The model does not use external data to represent time-varying processes. All inputs are generated within the ABM itself.

## 2.7 Submodels

The following subsection describes in detail the search procedures and models of (mis)communication that in combination constitute the submodels we explore.

The first search procedure is a *relay* model in which the members of a group tackle the hill climbing problem sequentially. That is, the first member of a group utilizes her own search heuristic to explore the landscape on her own, and then reports the results of her search (i.e., the location of the highest point she discovered) to a second member of the group who picks up the search where the first agent left off utilizing the second agent's (possibly different) search heuristic. The second member then passes the search off to the third, and so on, with the last group member passing things back to the first member. This process is repeated until no group member is able to improve upon the location from which their individual search commences.

The second procedure we consider has what we call a cooperative *tournament* structure in which the members of a group engage in search simultaneously. On this model members of a group each utilize their respective heuristics to explore the landscape simultaneously. After each has exhausted their individual search, they then report the results to the group, and the entire group then moves to the reported location of the highest peak identified in that round of search. The respective members of the group then begin search anew from this location, and the process repeats itself until no member of the group is able to identify a higher location than the one from which they began a round of search.

Third, we consider a *hybrid* procedure in which groups divide themselves into subgroups that work together using a cooperative tournament approach, but where each subgroup engages in separate relay searches, the results of which are then used as inputs for the cooperative tournament approach. In other words, the group comprised of the nine agents {a, b, c, d, e, f, g, h, i} might divide itself into three subgroups: $\alpha = \{a, b, c\}$; $\beta = \{d, e, f\}$; $\gamma = \{g, h, i\}$. The members of $\alpha$ (a, b, and c) would then engage in relay search with one another, as would the members of $\beta$ and $\gamma$. Each subgroup would then report the results of its search to the other subgroups, and they would all move to the location of the highest reported peak from which each subgroup would begin a new relay search. This procedure would repeat until no subgroup could improve upon the location from which a round of search began.

The first two search procedures replicate the procedures utilized in [2]. That paper reports that team dynamics did not have an impact on the finding that diverse groups outperform groups of experts. Because we anticipated the team search dynamics having an effect on the impact of miscommunication we thought it worth exploring the third hybrid procedure. Also, note that because team search dynamics affect how groups communicate it is also likely to impact the way in which miscommunication might be linked to diversity. To account for this we focus on the limit cases. Specifically, we assume that when agents in a team (or subgroup) are utilizing the relay procedure miscommunication is a function of the similarity of the heuristics of whatever agent has just completed searching and the agent to whom she is passing off the search. And when teams are utilizing the cooperative tournament procedure miscommunication is a function of the maximum measure of dissimilarity between any two members of the team.

We consider five ways of introducing communication and the possibility of miscommunication into the model.

i) The first is a model of perfect communication in which communication between agents is lossless. This model serves two purposes. First, it allows us to test whether our model is able to replicate the results reported in Hong and Page and Singer. Second, and more importantly, it provides us a baseline against which to compare the impact of the models of

miscommunication we consider.

To model miscommunication, we assume that when agents report the results of their individual search efforts to fellow team members the results are reported with error. We consider four models of the error induced by miscommunication. To account for the possibility that diversity and miscommunication are linked some, but not all of these, assume that the degree of miscommunication is a function of how similar the members of a respective group are. The four models we consider are:

ii) A fixed error model which assumes that miscommunication induces an error in the reported location of the highest peak identified in a round of search equal to half of the measure of dissimilarity between the agents. When we utilize the Hamming distance to measure similarity this number varies between 0 and 1.5 units along the ring-shaped landscape since the maximum Hamming distance between agents with the 3-tuple heuristics we consider is 3. On the other hand, when we utilize Manhattan distance the maximum induced error grows considerably larger as a function of how big the moves included in a heuristic are allowed to be (this is because the maximum Manhattan distance between heuristics is equal to $n \times k$ where n is the number of components in a heuristic, and k is the maximum size of a jump).

iii.a) A stochastic model in which the reported location is drawn from a normal distribution with a mean equal to the true location of the highest peak identified in a round of search, and a standard deviation that is a function of the similarity measure we utilize. Specifically, $\sigma = \sqrt{Similarity\ Measure}$. This allows us to accommodate the plausible hypothesis that the error induced by miscommunication is sensitive to the differences between agents, but is not fixed.

iii.b) An alternative stochastic model in which the reported location is drawn from a Poisson distribution with a mean also set at the true location of the highest peak identified in a round of search. This provides us with a plausible representation of an error generating process that is not sensitive to the differences between agents, but which has roughly the same shape as a normal distribution, allowing us to explore the role miscommunication plays in driving our results independently of its relationship to diversity.

iii.c) A third stochastic model in which the reported location is drawn from an exponential distribution with mean located at the true location of the highest peak, which is also insensitive to differences between agents. This model allows us to explore the possibility that the error induced by miscommunication is biased which, among other things, may impact how the error is propagated.

To clarify how miscommunication works, consider a simple example of two agents searching the landscape together. If the heuristics of Agent 1 and Agent 2 are, respectively, $H_1 = (1, 2, 3)$ and $H_2 = (10, 1, 7)$. Then, $d_{man}(H_1, H_2) = 9 + 1 + 4 = 14$ and $d_{han}(H_1, H_2) = 1 + 1 + 1 = 3$.

Suppose the agents start at spot 0 and search according to the relay method. Agent 1, let us suppose, finds the peak at position 2 and reports her result to Agent 2. This is where miscommunication enters.

Assume we use the Hamming measure of diversity. Then, according to communication model (ii), i.e. the "fixed model," Agent 2 wouldn't hear "position 2," but would hear, instead, "position 3.5." Rounding, Agent 2 would commence his search from position 4, rather than 2. If instead we measure diversity according to the Manhattan distance, then Agent 2 would hear "position 9."

Alternatively, if miscommunication occurs according to model (iii.a), the error would be randomly generated according to a normal distribution. The mean location of this distribution would be $\mu = 2$, and its standard deviation would be $\sigma = \sqrt{3}$ if we utilize Hamming distance, and $\sigma = \sqrt{14}$ if we utilize Manhattan distance. Rather than hearing "position 2," then, Agent 2 would hear "position 2+r," where r is a random error term, drawn from the relevant normal distribution.

On the other hand, if the miscommunication occurs according to models (iii.b) or (iii.c), then it is insensitive to the degree of diversity. Instead, the error is generated according to a Poisson or exponential distribution, respectively. In this case, rather than hearing "position 2," Agent 2 hears "position x," where x ~ Pois(2), in model (iii.b), and x ~ Exp(½) in model (iii.c).

When there are more than two agents in a team, things unfold similarly with Agent 2 reporting her final position to Agent 3 with some error, Agent 3 reporting hers with some error to Agent 4, and so on, with Agent n reporting her final position with error to Agent 1 who would then recommence searching if this reported position is different than the point at which she had previously stopped.

The tournament search method unfolds in an analogous manner with agents' positions at the end of a round of search being reported with error. However, because agents search simultaneously and report their positions to the entire team, as opposed to searching sequentially and communicating in pairs, miscommunication is introduced in a slightly different manner. Specifically, tournament search begins with each agent in the team beginning their search from the same location using their individual search heuristic. A round of search is complete once all the agents get stuck, at which point the highest peak located in that round of search is identified. That location is then reported with error to the entire team. In models (iii.b) and (iii.c), where miscommunication is uncorrelated with diversity, these error-ridden locations are once again drawn from the relevant Poisson and exponential distributions. The entire team recommences search from this new location, with the process repeating until no agent can improve upon the location from which a round of search begins.

Models (ii) and (iii.a), where miscommunication is a function of how different the members of a team are from one another, introduce additional complexities. Specifically, we must introduce a measure of the diversity of a team with which the error induced by miscommunication is correlated. We focus on the limit case and define the relevant error as a function of the maximum measure of difference between any two team members. To illustrate, consider the two agents introduced above characterized by the heuristics $H_1 = (1, 2, 3)$ and $H_2 = (10, 1, 7)$, and a third agent characterized by $H_3 = (4, 5, 6)$. As we saw above $d_{man}(H_1, H_2) = 14$, while the distance between Agents 1 and 3 and Agents 2 and 3 are, respectively, $d_{man}(H_1, H_3) = 3 + 3 + 3 = 9$ and $d_{man}(H_2, H_3) = 6 + 4 + 1 = 11$. The maximum difference between two team members in this case is between Agents 1 and 2. If the highest peak located in a round of search was at "position 2", as it was in the previous example, then this position would be reported as it was in that case (i.e for model (i) it would be reported not as "position 2" but as "position 9" where this corresponds to "$2 + (\max_{1 \leq x \leq 3} d_{man}(H_i, Hj))/2$", and for model (ii) this would be "position $2 + r$" where $r \sim N(2,14)$). When Hamming distance is instead being utilized as the diversity measure it is simply substituted for the Manhattan distance here, noting, however, that the maximum Hamming distance characterizing a team's diversity may be associated with a different pair of agents.

Two additional aspects of how search is implemented in the model warrant comment. The first concerns a difference in how tournament search operates with and without miscommunication. When communication is lossless the agent who identifies the highest peak in a round of search is effectively prevented from contributing to the next round of a team's search. When

miscommunication is possible, however, this is no longer the case. This is because the introduction of error entails that the *actual* "best location" identified in a round of search will be different from the *reported* "best location" from which agents begin the subsequent round of search. Notably, this way of operationalizing miscommunication lends itself to a natural interpretation, viz. that because of communication frictions the agent who identifies the "best location" in a round of search will not necessarily know that she had done so.

The second aspect of search that requires further comment concerns the way miscommunication manifests itself in the hybrid search procedure we consider. That procedure divides teams into subgroups. Within subgroups agents engage in relay search with one another, while subgroups as a whole engage in tournament search. Accordingly, miscommunication is introduced in two ways. Within subgroups error is generated as it is in the relay model with each baton passing between agents occurring with error. Every time a subgroup completes a round of search, though, additional error is generated as we described above for the tournament model. Specifically, the "best location" identified by a subgroup is reported to the other subgroups with error. In both of these cases error is modeled in four ways, and when models (ii) and (iii.a) are utilized the relevant measure of diversity that error is correlated with is the one utilized in the relay and tournament methods, respectively (i.e., the distance between the agents engaged in a baton passing in the relay case, and the maximum distance between any two agents in the tournament case).

In sum, the submodels described above allow us to accommodate two different ways of calculating similarity between agents, three group search dynamics, and four ways of modeling the error induced by miscommunication.

## 3. Summary of results

Accounting for the possibility of miscommunication does not undermine the finding that diversity can be beneficial to groups engaged in rugged landscape optimization problems. However, our analysis suggests that the extent to which diversity is beneficial, and the circumstances in which a group's diversity is likely to trump the individual ability of its members are highly dependent upon context. Below is a summary of the most important findings of our analysis. As described in 2.4 above, the analysis was conducted on a data set built from 8400 runs of the model in which each parameterization of the model was run 50 times. To facilitate comparison with Hong and Page and Singer the analysis summarized here focuses on the 2100 runs for which the maximum heuristic component was 20 (the same parameterization they report results for). Paired *t*-tests were utilized to compare expert and random team performance for each submodel characterized by a combination of a search method, error model, and diversity measure, and all of the results reported here are significant at the $p < .05$ level. The figures that follow provide a representative overview of the data on which this analysis is based. Complete data can be found in S1 Data.

- *As the landscape becomes smoother, diverse groups and expert groups tend to converge in their performance.* This essentially replicates the primary findings supporting the DTA result. When landscapes are sufficiently rugged random teams tend to outperform teams comprised of experts because the random teams are more functionally diverse. However, as we would expect, the relative benefit of diversity diminishes as landscapes become less rugged, and disappears entirely when landscapes are relatively smooth. Similarly, as the size of the maximum jump that can be included in a heuristic gets smaller, the relative benefit of diversity diminishes, as do the magnitude of the differences between diverse and expert groups. This is evident in both panels of Fig 1, where the blue lines represent the difference between random groups and expert groups with perfect communication. On rugged landscapes, most of

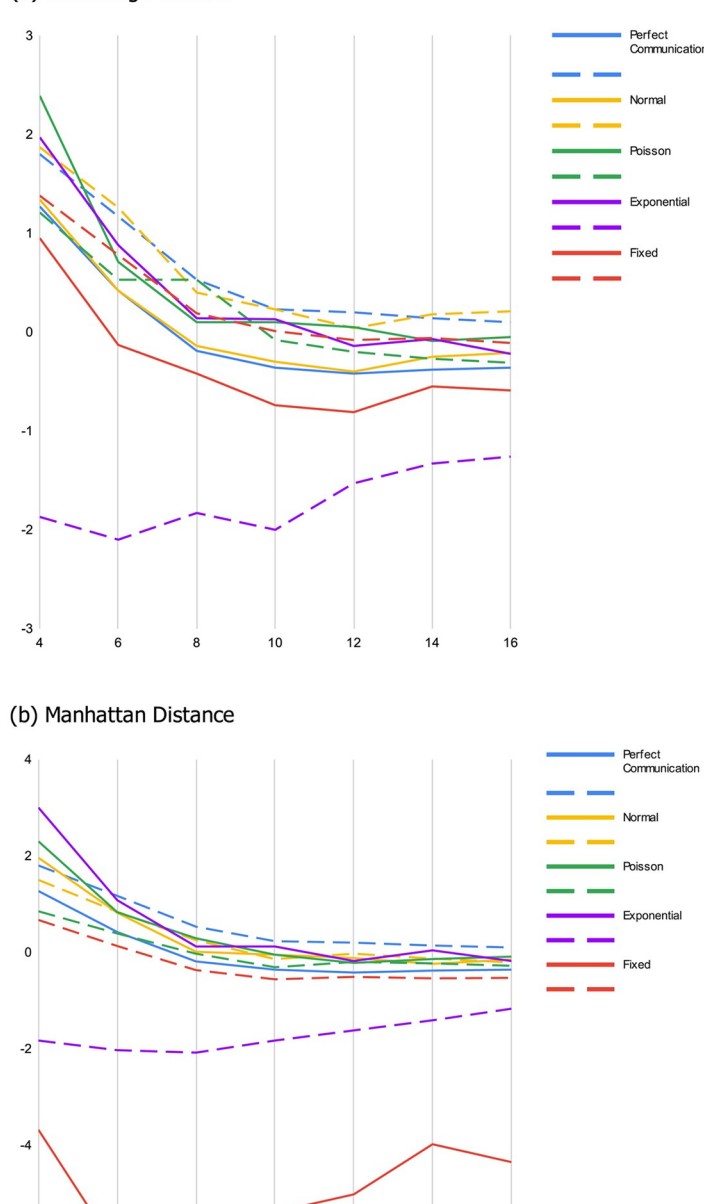

**Fig 1. Difference between random and expert performance for relay (solid) and tournament (dashed) search methods.**

these lines are significantly above 0, indicating a substantial advantage for the more diverse random groups. This advantage wanes as landscapes become smoother.

- *When miscommunication is possible the benefit of diversity is more nuanced.* Although diversity continues to be relatively more beneficial when landscapes are more rugged (as one

would expect), whether diverse teams outperform teams of experts and to what extent both become sensitive to the interaction between the four primary parameters of the model: i) the degree of ruggedness of the landscape, ii) the nature of the error induced by miscommunication, iii) the way groups work together, and iv) how we measure diversity.

- *With some exceptions, the more rugged the landscape, the more beneficial diversity is.* Diverse teams tend to perform better than expert teams when the smoothing parameter is set to a low value. This advantage is lost as landscapes become increasingly smooth. An interesting exception occurs when miscommunication follows an exponential distribution and teams search using the cooperative tournament method (Fig 1). In a handful of parameter combinations, diverse groups begin with a disadvantage, made worse by increasing the smoothness of the landscape. This is especially apparent in Figs 1b and 3b.

- *Fixed error has a significant and negative effect on diverse teams.* The magnitude of this effect is also dependent on how we measure diversity, with the Manhattan metric having a significantly bigger impact. To see this, compare panels (a) and (b) in Fig 1. This effect is not surprising given that the Manhattan metric can take on larger values and the error functions we utilize are sensitive to the magnitude of our measure of diversity. Perhaps more interesting, the relay search method tends to exacerbate this effect, as is apparent when comparing the dashed green line to the solid green line in both panels.

- *When the error induced by miscommunication is biased but not a function of how diverse groups are, the group search dynamic matters a lot.* This is true of the Poisson distributed error, but is especially apparent when reported locations are drawn from an exponential distribution. Diverse teams do better when utilizing the relay method, and expert teams do better when using the cooperative tournament method. This result obtains for all but the smoothest landscapes and regardless of the measure of diversity we use. This result is one of the most striking features in Fig 1a and 1b, where the dashed orange line (exponential with tournament search) lies far below the orange solid line (exponential with relay search).

- *Surprisingly, when teams utilize relay search, miscommunication can sometimes improve the performance of diverse teams relative to expert teams.* There are three cases in which we observe this effect: (i) when error is a normally distributed function of how diverse groups are, and we measure diversity with Manhattan distance (Fig 2b), (ii) when error is drawn from a Poisson distribution (Fig 2a and 2b), and (iii) when error is drawn from an exponential distribution (Fig 2a and 2b). Notably, case (i) is the only exception to the general finding that the benefit of diversity tends to diminish when miscommunication is a function of how different the members of a group are from one another. In contrast to the cases where miscommunication is detrimental, however, there is no obvious mechanism that explains this result.

- *When miscommunication is a function of diversity, the search method impacts whether the group score is a monotonic function of both diversity and ruggedness* (and when it is not it also impacts where the inflection point in the relationship is). These non-monotonic relationships exist most apparently in searches involving fixed miscommunication with hybrid search methods (see Fig 3), as well as the relay search when diversity is measured using the Manhattan metric (Fig 1b).

- *When miscommunication is a function of diversity, the diversity measure also impacts whether the group score is a monotonic function of both diversity and ruggedness.* For example, when miscommunication follows a normal distribution, random score (minus expert score) changes non-monotonically in smoothness when diversity is measured using the Hamming

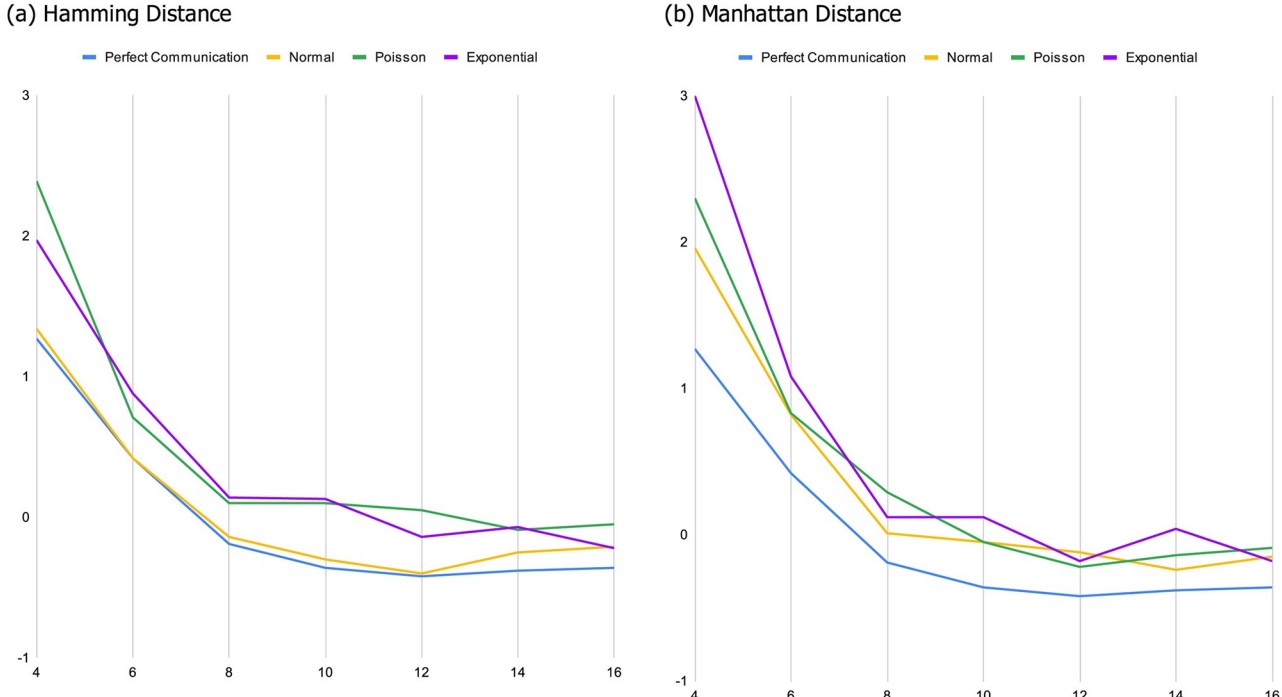

**Fig 2. Difference between random and expert performance for relay search methods with a focus on the effect of randomly distributed communication errors.**

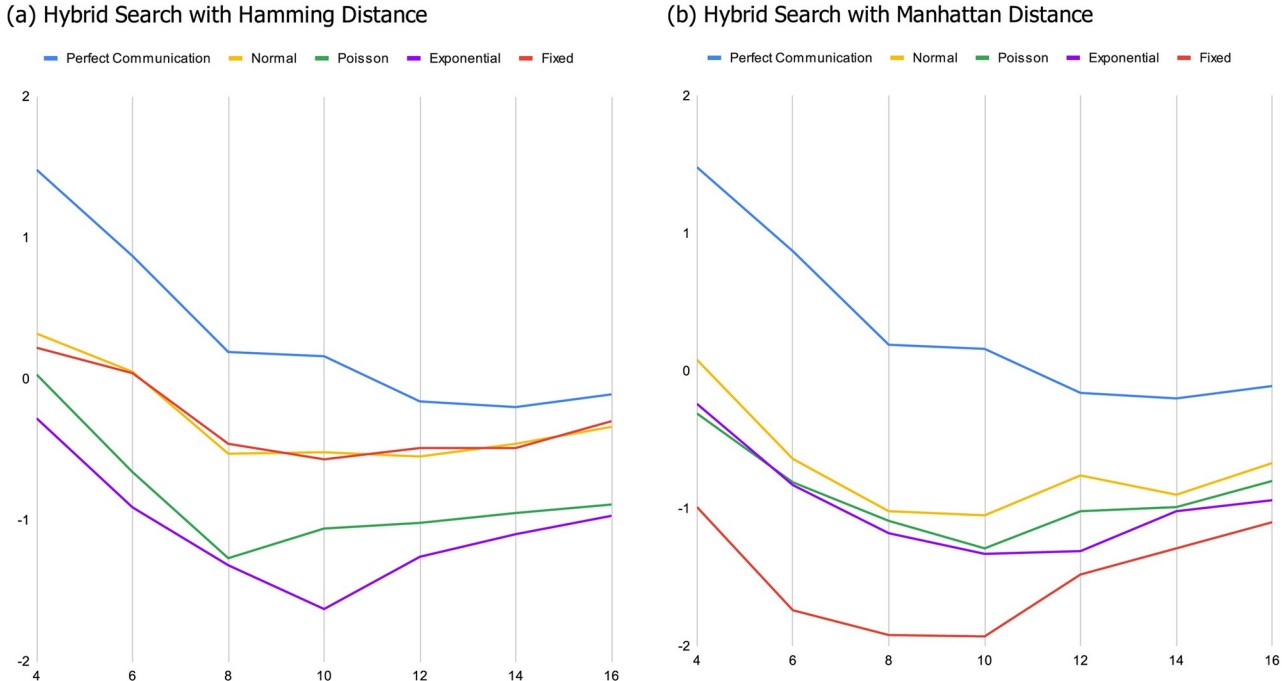

**Fig 3. Difference between random and expert performance for hybrid search method.**

metric and decreases monotonically when using the Manhattan metric. This can be seen by comparing Fig 1a and 1b.

Each figure depicts the relative performance of random and expert groups on landscapes of various degrees of ruggedness. The mean difference between random and expert group performance calculated from 50 simulations on randomly seeded landscapes is reported on the y-axis (positive values correspond to cases where random groups outperform experts), and the measure of landscape ruggedness used to seed the landscapes is depicted on the x-axis (small values correspond to more rugged landscapes). Points on the landscape take on values between 0 and 100, so a mean difference of 2 corresponds to random groups performing about 2% better than expert groups on average.

## 4. Discussion

The results described above provide further evidence for the view that perspectival diversity can be an important tool for groups confronting rugged landscape optimization problems. However, motivating our work was the hypothesis that diversity is likely to be associated with interpersonal frictions like miscommunication. By exploring several ways of modeling miscommunication the simulations described above provide clear evidence that these frictions are likely to moderate the ability of groups to leverage the benefits of diversity. Sometimes this effect is quite significant, and, more importantly, this is true even when miscommunication is not itself a function of diversity. Together these findings motivate:

> Lesson 1: *Scholars interested in studying the impact of diversity on teams, and organizations hoping to reap the benefits of diversity, must account for the interpersonal frictions that individuals working together inevitably confront, especially (but not only) when there is reason to believe that these frictions could be associated with diversity.*

Reflecting on the role miscommunication plays in our model illustrates why this is important. As we discussed earlier the computational findings underlying the diversity trumps ability postulate are driven by the fact that there is too much overlap in the search heuristics employed by the members of expert teams. As a result, experts tend to explore less terrain and get stuck in the same places. What our simulations illustrate, however, is that miscommunication can actually be beneficial to expert teams because it provides a mechanism for preventing them from prematurely converging on a local optimum that could be improved upon with more extensive search. For random teams, on the other hand, because they already have the functional diversity that prevents premature convergence, the presence of miscommunication simply introduces additional noise that detracts from the ability of group members to successfully build upon each other's efforts.

The importance of accounting for interpersonal frictions when studying team dynamics is not the only lesson we want to emphasize, though. The simulations we report on above suggest that there are significant interactions between team search dynamics and the various ways we model miscommunication. Whether diverse groups outperformed expert groups in the presence of miscommunication (and by how much) was a function of both the search procedure a teams utilized and the assumptions we made about what drove the miscommunication and how it impacted the sharing of information among team members. One way of thinking about this is to say that the way a team works together has a significant impact on how interpersonal frictions impact team performance. Alternatively, we can say that the nature of the frictions that arise between individuals has a significant, but varied impact on

how fruitful various team dynamics are. Together these motivate the primary takeaway of our analysis:

Lesson 2: *Institutional dynamics mediate the impact of interpersonal dynamics*

*(and vice versa).*

With respect to the impact of diversity on teams, this means that: to leverage the functional diversity of its members it's critical that a group or organization employ procedures that are suited to the characteristics of the individuals that comprise the team and the interpersonal frictions that arise between them.

Once again, reflecting on our model can illustrate why the lesson described above is important. By far the most significant result of our simulations was that, when miscommunication is modeled as a fixed error that is a function of how different the members of a team are, the impact of miscommunication on diverse teams is significant and negative. This negative impact is much larger when teams utilize the relay method of search, though, so much so that when they utilize relay search diverse teams significantly underperform expert teams on landscapes of all degrees of ruggedness. This result is driven by two things. First, the assumption that miscommunication induces fixed error that is a function of diversity presents a worst-case scenario for diverse teams relative to expert teams because they face much larger error, and this error afflicts all their interactions. Second, the relay search method then propagates (and compounds) these errors, because each team member inherits error at the start of their individual search and then introduces additional error at the end of it. Members of teams employing the cooperative tournament approach, on the other hand, all start their search from the same location, and, although that location differs from the true location that a previous round of search ended at, because the agents all search simultaneously and in different ways, the error is less likely to propagate. Furthermore, additional error is only introduced once after all members have completed a round of search and they have identified a new best location (as opposed to new error being introduced after each team member's individual contribution).

It should be emphasized that the two lessons we've highlighted are not entirely surprising. Both reflect the Coasean insights that: i) institutions are needed in part because they help us deal with transaction costs, and ii) optimal institutions are shaped by the nature of those transaction costs [51, 52]. Our work builds upon these insights in two ways. First, it provides evidence that mechanisms that help groups capture the benefits of diversity in some contexts, are not guaranteed (or even particularly likely) to work when they are adopted in other contexts. Indeed, translating some mechanisms from one context to another may even be counterproductive. Second, it provides a framework for understanding why this is the case, and, although the framework is relatively abstract, the key components of the model–i) rugged landscape optimization, ii) constraints on individual search, iii) various group search strategies, and iv) interpersonal frictions–are common characteristics of problems that arise in numerous domains. Indeed, the lessons we've drawn here are consistent with the widespread finding in the empirical literature on diversity and teams that diversity is often beneficial in some respects and detrimental in others, and the particular friction we've incorporated into our model–miscommunication–can arguably help explain some of these results. For example: that cognitive diversity is good for divergent creativity (associated with generating ideas), but can inhibit convergent creativity (associated with the ability to build upon and integrate ideas) [53–55]; that linguistic and ethnic diversity have a positive effect on groups when individuals have to collaborate on preventative tasks, individual contributions are hard to disentangle, and external threats are easily identifiable, but can have a negative (and more significant) impact when

these conditions don't obtain [56]; or that diverse teams identify and utilize more task relevant information in ways that improve team performance, but often as a result of experiencing more conflict that has a detrimental impact on perceived performance and the desire to continue working together [16, 57].

## 5. Limitations and directions for further research

As we noted above our model is both abstract and simple. This has both advantages and disadvantages. The chief advantages are i) that the results of the model are relatively easy to interpret, and ii) because the key components of the model are characteristic of problems that arise in myriad domains, the results are broadly generalizable. As we noted above, however, the primary takeaway of our analysis was that the impact of diversity on groups is highly sensitive to characteristics of i) the individuals comprising the group, ii) the way they work together, and iii) the nature of the problem they confront. This points to the chief disadvantage of analyzing a relatively simple model, namely that we abstract away from many other characteristics of groups and problems that likely mediate the impact of diversity. Two simplifications, in particular, stand out.

First, miscommunication is not the only example of an interpersonal friction that might impact how groups work together. As we suggested previously, differing epistemic standards, and different values/goals are likely to give rise to similarly significant frictions, and as with miscommunication there is reason to believe that both are likely to be associated with diversity. Furthermore, these disparate sources of interpersonal frictions are also likely to interact with one another.

Second, our model only incorporates a single type of diversity. Although we consider multiple measures of how different the members of a team are from one another, we model diversity entirely in terms of the heuristics that individual agents use to explore the landscape. This provides a useful analog for how individuals solve problems, but that is only one aspect of cognitive or perspectival diversity. An important question in diversity research is how various aspects of diversity map on to the sort of cognitive diversity we consider here. Also important is whether things like demographic diversity give rise to the sort of frictions we consider here (or impact groups in other ways) that are independent of the role they play in generating cognitive diversity.

While there are several other technical simplifications worth discussing, the two mentioned above suggest promising lines for further research. One such line involves investigating additional impacts of diversity-induced frictions. Some of these frictions, such as decreased cooperativeness or the lack of a commonly shared characterization of the problem, will likely decrease the efficacy of diverse teams. However, other frictions—such as the increased need to clearly communicate one's reasoning or the motivation to exert greater effort in justifying one's solutions—may enhance the effectiveness of diverse teams. Which force predominates and whether diversity acts as a hindrance or a benefit may well depend upon our exact understanding of diversity. If team members disagree sharply about the problem itself, this may thwart their efforts to agree upon the best solutions. On the other hand, modest disagreements about the problem combined with different approaches to solving it, may induce greater effort in searching and justifying the results of one's search, ultimately leading to superior solutions.

The results presented here also raise new questions in the context of parallel research projects that highlight different aspects of the search dynamics of problem-solving teams. For example, Frigotto and Rossi (2012) analyzes the effects of diversity and communication in the context of two-person teams attempting to select an explanatory theory [46]. They use their model to assess whether or not the intensity of communication enhances or undermines the

quality of theory selection. They find, interestingly, a non-monotonic relationship between communication intensity and performance; at first, increasing levels of communication enhance search, but, beyond a certain point, increasingly intense communication can confuse agents and undermine their attempts to select a good theory. Our model suggests that it is not merely the intensity of communication that matters. We must also know (1) whether that communication contains errors, and (2) what search method is employed by the group. Future work should examine how these two variables interact with the main variable studied by Frigotto and Rossi, i.e., communication intensity.

In a different vein, there is a significant and growing body of literature on how social networks affect cooperative problem-solving. For instance, Fontanari and Rodrigues (2016) explores how the interaction between landscape topology and communication networks impacts problem-solving in NK landscapes [39]. They find that the size of a group significantly impacts the optimal degree of connectivity in settings where agents learn via mistaken prone imitation. Specifically, they show that higher degrees of connectivity are often bad for groups as they get larger because the increased levels of imitation this facilitates reduces the functional diversity of the group. This finding is related to our own insofar as the search methods we consider exhibit various degrees of connectivity (with teams employing the tournament method exhibiting greater connectivity than those employing relay methods). However, Fontanari and Rodrigues do not consider the possibility that error induced by imitation might take different functional forms and/or might be correlated with a measure of how different (or distant) agents are from one another. Our work suggests that extensions of their model that try to accommodate these facts would be valuable.

Similarly, there are now numerous studies that explore how social influence impacts information propagation in groups, and how this affects collective problem-solving [58]. For example, there are many contexts in which social influence undermines the wisdom of the crowds [59, 60]. Given the connection between the wisdom of crowds result and the diversity trumps ability theorem, and given that the diversity of a group plausibly impacts how influential its members are vis-à-vis other members it stands to reason that the results described here have implications for social influence models and vice versa. Indeed, it's worth emphasizing that many models of social influence simply treat influence as a function of connectivity in a network. But there is room to enrich the picture of why some agents are more influential than others (and who they influence) beyond the location of an agent in a network, and measures of how different agents are from one another provide one such mechanism for enriching these models [61, 62]. A promising frontier for future work would be to develop models that do this that are validated by complementary line of empirical work [58]. For the purposes of this paper, however, the benefits of incorporating a mechanism for varying the influence of agents that might also have been correlated with diversity were outweighed by the detrimental effect that would likely have on our ability to address our core concern, viz. how the introduction of communication frictions complicate interpretations of the DTA theorem.

Finally, we want to emphasize again that many of the benefits associated with diversity discussed here are a function of the fact that diverse groups tend to engage in more extensive search. More extensive search allows us to identify better solutions to problems, but it is important to remember that effort is costly and more extensive search typically requires more effort. At the very least, more extensive search typically involves a greater time investment. This issue calls for a new measure that seeks to assess the efficiency, not merely the absolute value, of a team's performance. Such a measure would involve looking at the score per unit of effort or time. Future work should include such a measure to better assess the advantages of diversity in teams.

In conclusion, our analysis suggests that many of the conclusions drawn from previous work on diversity must be qualified, and this is especially true of claims that appeal to the "Diversity Trumps Ability Theorem". Diversity is a resource that groups can utilize. But doing so requires having the right institutions in place. While the Hong-Page model provides a useful framework for analyzing the impact of diversity on groups, future work should focus on extensions of that model (and others like it) that better capture the way group dynamics interact with interpersonal frictions to augment/diminish the costs and benefits associated with diversity.

## Supporting information

**S1 File. Manhattan distance measure.**
(DOCX)

**S2 File.**
(nlogo)

**S1 Data.**
(XLSX)

**S1 Table. HP-miscommunication data analysis summary.**
(PDF)

## Acknowledgments

We are grateful to Daniel Singer for sharing the code for the baseline ABM that we extended, to Ryan Yonk for feedback on our data analysis, and Gabe McFadden for helping us with data visualization. We would also like to thank colleagues at Chapman University, especially Drew Moshier, John Thrasher, and Erik Kimbrough, and to audiences at Cal State Sacramento, the PPE Society, and the Midwest Political Science Association where various versions of this paper were presented.

## Author Contributions

**Conceptualization:** Keith Hankins, Ryan Muldoon, Alexander Schaefer.

**Data curation:** Ryan Muldoon, Alexander Schaefer.

**Formal analysis:** Keith Hankins, Ryan Muldoon, Alexander Schaefer.

**Methodology:** Ryan Muldoon.

**Software:** Ryan Muldoon.

**Visualization:** Alexander Schaefer.

**Writing – original draft:** Keith Hankins.

**Writing – review & editing:** Keith Hankins, Ryan Muldoon, Alexander Schaefer.

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
