## [Decision Letter · Decision Letter 0]

14 Aug 2022

PONE-D-22-19336Does (mis)communication mitigate the upshot of diversity?PLOS ONE

Dear Dr. Hankins,

Thank you for submitting your manuscript to PLOS ONE. After careful consideration, we feel that it has merit but does not fully meet PLOS ONE’s publication criteria as it currently stands. Therefore, we invite you to submit a revised version of the manuscript that addresses the points raised during the review process.

Your paper has been revised by two expert reviewers. They highlighted significant changes to be made to improve your paper. You are therefore invited to make the best use of those suggestions to provide a revised version of your paper.  You are kindly requested to also check the website for possible reviewer attachment(s).==============================

We look forward to receiving your revised manuscript.

Kind regards,

Pierluigi Vellucci

Academic Editor

PLOS ONE

Journal Requirements:

Reviewers' comments:

Reviewer's Responses to Questions

**Comments to the Author**

1. Is the manuscript technically sound, and do the data support the conclusions?

Reviewer #1: Yes

Reviewer #2: Partly

2. Has the statistical analysis been performed appropriately and rigorously? 

Reviewer #1: Yes

Reviewer #2: Yes

3. Have the authors made all data underlying the findings in their manuscript fully available?

Reviewer #1: Yes

Reviewer #2: Yes

4. Is the manuscript presented in an intelligible fashion and written in standard English?

Reviewer #1: Yes

Reviewer #2: Yes

5. Review Comments to the Author

Reviewer #1: The goal of the paper is to include communication between agents into a model of problem solving by teams of diverse agents, whereby communication might occur with error, and the error might be correlated with agent diver- sity. The authors clarify the moderating roles of task complexity, the number of problem solving approaches, the ways of conceptualizing a problem, and insti- tutional characteristics (in terms of search processes). I think the paper covers an interesting and important topic and is of interest to the diverse PLOS One readership. However, I also think that the authors should clarify some issues (in particular related to the model). I will be happy to read the rebuttals to my comments and reconsider my view on the manuscript.

Detailed comments are provided in the attached referee report.

Reviewer #2: This paper advances the Diversity Trumps Ability theorem by introducing communications between solvers in the underlying model. The paper is interesting and, in my opinion, makes the contribution to the theory of problem solving. However, there are some points that should be resolved before the paper can be accepted to publication.

Let me start with the most important comments

1) First, a simple search procedure using the GoogleScholar (I opened the papers that cited the seminal article [Hong L, Page S. Groups of diverse problem solvers can outperform groups of high-ability problem solvers. Proc Natl Acad Sci USA. 2004 Nov 16;101(46):16385–9.] and then went through the ten top ones) ends up with finding the following paper:

[Frigotto, M.L., Rossi, A. Diversity and Communication in Teams: Improving Problem-Solving or Creating Confusion?. Group Decis Negot 21, 791–820 (2012). https://doi.org/10.1007/s10726-011-9250-x ]

I did not go too deep in this paper but I suppose that it has some common features with the manuscript under review. On this occasion, I assume that the authors should put more efforts on making the embedding of their results.

2) As you add communication into the model, you should account for the fact that interacting agents not only share information but may also influence each other - see the perfect review article on social influence models [Flache, A., Mäs, M., Feliciani, T., Chattoe-Brown, E., Deffuant, G., Huet, S., & Lorenz, J. (2017). Models of social influence: Towards the next frontiers. Journal of Artificial Societies and Social Simulation, 20(4).]

By saying "agents may influence each other," I mean that agents can mimic other agents' strategies (especially, if they are advantageous).

As within your approach, the influence may also depend on how diverse the interacting agents are in the terms of opinions - in your case, opinions are strategies (see, for example, [Takács, K., Flache, A., & Mäs, M. (2016). Discrepancy and disliking do not induce negative opinion shifts. PloS one, 11(6), e0157948.] and [Kozitsin, I. V. (2021). Opinion dynamics of online social network users: a micro-level analysis. The Journal of Mathematical Sociology, 1-41.])

I suppose that you should include the social influence mechanism in your model, with some parameter representing the power of the influence effect (the current results can be understood as those that correspond to the case of the zero value of the parameter).

Note that social influence can substantially affect groups' decisions (this result was obtained within the line of research that studied the wisdom of crowd phenomenon (which is close, in my opinion, to the DTA theorem) - see [Lorenz, J., Rauhut, H., Schweitzer, F., & Helbing, D. (2011). How social influence can undermine the wisdom of crowd effect. Proceedings of the national academy of sciences, 108(22), 9020-9025.])

Minor comments

1) Implementing miscommunication error

I would like to see in Subsection 2.2 an illustrative example that demonstrates how the error depends on the diversity across the four models you introduced.

2) What I would like to see in the manuscript is an example of how the NK model landscapes are organized, with some illustrations on agents' strategies. I was slightly confused with your example of agents' strategies (line 116), because you announce that the landscape is two-dimensional but the strategies imply only one dimension (left-right). Then, you speak about the ring topology (line 286). Maybe, by saying two-dimensional, you mean that the landscape is parametrized by two quantities - "the size and ruggedness".

I suppose that you should provide a more detailed explanation on the organization of the landscape.

3) Lines 312-319. I suppose that your explanation regarding the differences in metrics is redundant and can be safely removed from the main text.

4) Lines 86-87. Subsections 1.1, 1.2, not Sections, I assume?

5) Line 101. ... any given point is the landscape is a ... "on" instead of "is"?

6) Quality of Figures is bad. I do not know whether the online version of the manuscript in the case of acceptance would feature a better quality, but for now it takes some efforts to investigate the differences between lines.

6. PLOS authors have the option to publish the peer review history of their article (what does this mean?). If published, this will include your full peer review and any attached files.

Reviewer #1: **Yes: **Stephan Leitner

Reviewer #2: No

---

## [Author Response · Author response to Decision Letter 0]

16 Feb 2023

Please see attached reply to reviewers.

---

## [Decision Letter · Decision Letter 1]

6 Mar 2023

Does (mis)communication mitigate the upshot of diversity?

PONE-D-22-19336R1

Dear Dr. Hankins,

We’re pleased to inform you that your manuscript has been judged scientifically suitable for publication and will be formally accepted for publication once it meets all outstanding technical requirements.

Kind regards,

Pierluigi Vellucci

Academic Editor

PLOS ONE

Additional Editor Comments (optional):

Reviewers' comments:

Reviewer's Responses to Questions

**Comments to the Author**

1. If the authors have adequately addressed your comments raised in a previous round of review and you feel that this manuscript is now acceptable for publication, you may indicate that here to bypass the “Comments to the Author” section, enter your conflict of interest statement in the “Confidential to Editor” section, and submit your "Accept" recommendation.

Reviewer #1: All comments have been addressed

Reviewer #2: All comments have been addressed

2. Is the manuscript technically sound, and do the data support the conclusions?

Reviewer #1: Yes

Reviewer #2: Yes

3. Has the statistical analysis been performed appropriately and rigorously? 

Reviewer #1: Yes

Reviewer #2: Yes

4. Have the authors made all data underlying the findings in their manuscript fully available?

Reviewer #1: Yes

Reviewer #2: Yes

5. Is the manuscript presented in an intelligible fashion and written in standard English?

Reviewer #1: Yes

Reviewer #2: Yes

6. Review Comments to the Author

Reviewer #1: The authors have addressed all of my previous comments appropriately, and the paper is now written in a clear and easy-to-understand manner.

I have a few minor comments:

1. In Section 5, I recommend adding "Conclusions" to the section title.

2. While Section 4 is labeled "Discussion," most of the discussion about existing work occurs in Section 5. Consider moving the discussion in Section 4 and limiting Section 5 to conclusions and limitations of the work.

3. On line 427, I believe the zero should be subscripted.

4. On line 113, is it always true that a dimension is a function of all n-1 other dimensions, or is this only the case in the extreme of full complexity?

Reviewer #2: My concerns have been addressed.

I like the improved version of the manuscript and I appreciate the effort the authors put to meet my and the other reviewer's reservations.

I have no other comments. Congratulation!

7. PLOS authors have the option to publish the peer review history of their article (what does this mean?). If published, this will include your full peer review and any attached files.

Reviewer #1: **Yes: **Stephan Leitner

Reviewer #2: No

---

## [Editor Report · Acceptance letter]

14 Mar 2023

PONE-D-22-19336R1 

Does (mis)communication mitigate the upshot of diversity? 

Dear Dr. Hankins:

I'm pleased to inform you that your manuscript has been deemed suitable for publication in PLOS ONE. Congratulations! Your manuscript is now with our production department. 

Kind regards, 

on behalf of

Dr. Pierluigi Vellucci 

Academic Editor

PLOS ONE